

# Analytically tractable climate-carbon cycle feedbacks under 21st century anthropogenic forcing

Steven J. Lade[1,2,3], Jonathan F. Donges[1,4], Ingo Fetzer[1,3], John M. Anderies[5], Christian Beer[3,6], Sarah E. Cornell[1], Thomas Gasser[7], Jon Norberg[1], Katherine Richardson[8], Johan Rockström[1], and Will Steffen[1,2]

[1]Stockholm Resilience Centre, Stockholm University, Stockholm, Sweden
[2]Fenner School of Environment and Society, The Australian National University, Canberra, Australian Capital Territory, Australia
[3]Bolin Centre for Climate Research, Stockholm University, Stockholm, Sweden
[4]Potsdam Institute for Climate Impact Research, Potsdam, Germany
[5]School of Sustainability and School of Human Evolution and Social Change, Arizona State University, Tempe, Arizona, USA
[6]Department of Environmental Science and Analytical Chemistry (ACES), Stockholm University, Stockholm, Sweden
[7]International Institute for Applied Systems Analysis, Laxenburg, Austria
[8]Center for Macroecology, Evolution, and Climate, University of Copenhagen, Natural History Museum of Denmark, Copenhagen, Denmark

*Correspondence to:* Steven Lade (steven.lade@su.se)

**Abstract.** Changes to climate-carbon cycle feedbacks may significantly affect the Earth System's response to greenhouse gas emissions. These feedbacks are usually analysed from numerical output of complex and arguably opaque Earth System Models (ESMs). Here, we construct a stylized global climate-carbon cycle model, test its output against complex ESMs, and investigate the strengths of its climate-carbon cycle feedbacks analytically. The analytical expressions we obtain aid understanding of carbon-cycle feedbacks and the operation of the carbon cycle. We use our results to analytically study the relative strengths of different climate-carbon cycle feedbacks and how they may change in the future, as well as to compare different feedback formalisms. Simple models such as that developed here also provide 'workbenches' for simple but mechanistically based explorations of Earth system processes, such as interactions and feedbacks between the Planetary Boundaries, that are currently too uncertain to be included in complex ESMs.

## 1 Introduction

The exchanges of carbon between the atmosphere and other components of the Earth system, collectively known as the carbon cycle, currently constitute important negative (dampening) feedbacks on the effect of anthropogenic carbon emissions on climate change. Carbon sinks in the land and the ocean each currently take up about one quarter of anthropogenic carbon emissions each year (Le Quéré et al., 2016). These feedbacks are expected to weaken in the future, amplifying the effect of anthropogenic carbon emissions on climate change (Ciais et al., 2013). The degree to which they will weaken, however, is highly uncertain, with Earth System Models predicting a wide range of land and ocean carbon uptakes even under identical atmospheric concentration or emission scenarios (Joos et al., 2013).



Here, we develop a stylised model of the global carbon cycle and its role in the climate system to explore the potential weakening of carbon cycle feedbacks on policy-relevant time scales (<100 years) up to the year 2100. Whereas complex Earth System Models (ESMs) are generally used for projections of climate, models of the Earth System of low complexity are useful for improving mechanistic understanding of Earth system processes and for enabling learning (Randers et al., 2016; Raupach, 2013). Compared to complex Earth System Models, our model has far fewer parameters, can be computed much more rapidly, can be more rapidly understood by both researchers and policy-makers, and is even sufficiently simple that analytical results about feedback strengths can be derived. Our stylised and mechanistically based climate-carbon cycle model also offers a workbench for investigating the influence of mechanisms that are at present too uncertain, poorly defined or computationally intensive to include in current Earth System Models. Such stylised models are valuable for exploring the uncertain, but potentially highly impactful Earth system dynamics such as interactions between climatic and social tipping elements (Lenton et al., 2008; Kriegler et al., 2009; Schellnhuber et al., 2016) and the planetary boundaries (Rockström et al., 2009; Steffen et al., 2015).

Analyses of climate-carbon cycle feedbacks conventionally distinguish four different feedbacks (Fig. 1) (Friedlingstein, 2015; Ciais et al., 2013). (i) In the land concentration-carbon feedback, higher atmospheric carbon concentration generally leads to increased carbon uptake due to the fertilisation effect, where increased $CO_2$ stimulates primary productivity. (ii) In the ocean concentration-carbon feedback, physical, chemical and biological processes interact to sink carbon. Atmospheric $CO_2$ dissolves and dissociates in the upper layer of the ocean, to be then transported deeper by physical and biological processes. The concentration-carbon feedbacks are generally negative, dampening the effects of anthropogenic emissions. (iii) In the land climate-carbon feedback, higher temperatures, along with other associated changes in climate, generally lead to decreased storage on land at the global scale, for example due to the increase in respiration rates with temperature. (iv) In the ocean climate-carbon feedback, higher temperatures generally lead to reduced carbon uptake by the ocean, for example due to decreasing solubility of $CO_2$. The climate-carbon feedbacks are generally positive, amplifying the effects of carbon emissions.

We begin by introducing our stylised carbon cycle model and testing its output against historical observations and future predictions of Earth System Models. Having thus established the model's performance, we introduce different formalisms used to quantify climate-carbon cycle feedbacks and describe how they can be computed both numerically and analytically from the model. We use our results to analytically study the relative strengths of different climate-carbon cycle feedbacks and how they may change in the future, as well as to compare different feedback formalisms. We conclude by speculating on how this stylised model could be used as a 'workbench' for studying a range of complex Earth system processes, especially those related to the biosphere.

## 2 Model formulation

There is a well-developed literature on stylized models used for gaining a deeper understanding of Earth system dynamics and even for successfully emulating the outputs of complex coupled atmosphere-ocean and carbon cycle models (Meinshausen et al., 2011a, c; Gasser et al., 2017a). Many such models are based on Budyko-Sellers (Budyko, 1969; Sellers, 1969) type

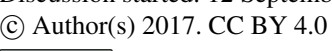



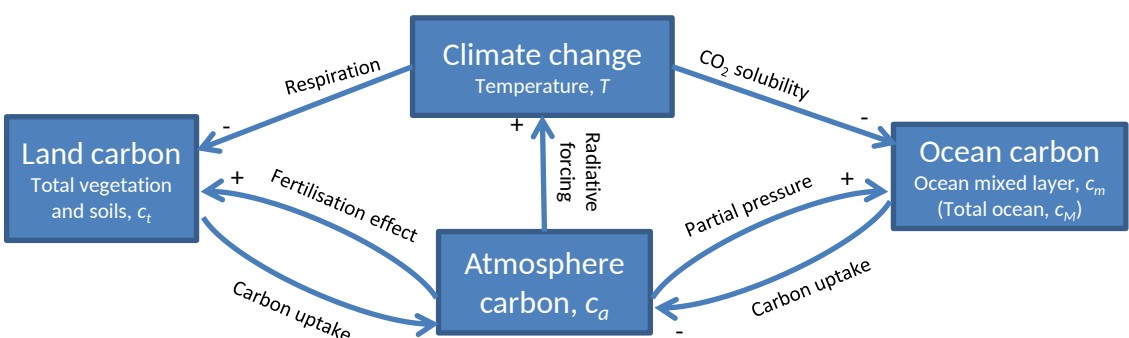

**Figure 1.** Climate-carbon cycle feedbacks and state variables as represented in the stylized model introduced in this paper. Carbon stored on land in vegetation and soils is aggregated into a single stock $c_t$. Ocean mixed layer carbon, $c_m$, is the only explicitly modelled ocean stock of carbon; though to estimate carbon-cycle feedbacks we also calculate total ocean carbon (Eq. (7)).

energy balance models and come in two flavors: models of mathematical interest motivated by the Earth system dynamics, and models focused on capturing essential features of the Earth system to reproduce broad empirical patterns. The former tend to focus on characterizing stability (e.g. Cahalan and North, 1979), and the existence of multiple equilibria given particular feedbacks (ice cap albedo) (e.g. North, 1990; Dıaz et al., 1997) or details of possible bifurcation structures Arcoya et al. (1998) in such models. Examples of the latter include studies of snowline stability (Mengel et al., 1988).

In the spirit of the energy-balance models described above, we constructed a global climate-carbon cycle model with the following characteristics:

1. The model includes processes relevant to the carbon cycle and its interaction with climate on the policy-relevant time scale of the present to the year 2100.

2. The model produces quantitatively plausible output for carbon stocks and temperature changes.

3. All parameters have a direct biophysical or biogeochemical interpretation, although these parameters may be at an aggregated scale (for example, a parameter for the net global fertilisation effect, rather than leaf physiological parameters). We avoid purely parametric fits to Earth System Models such as impulse response functions (Kamiuto, 1994; Gasser et al., 2017b; Joos et al., 1996; Harman et al., 2011).

4. The model is sufficiently simple that calculation of the model's feedback strengths is readily analytically tractable. This tractability may come at the expense of complexity, for example multiple terrestrial carbon compartments, or accuracy at millennial or longer time scales (Lenton, 2000; Randers et al., 2016).




Building on the work of Anderies et al. (2013), we constructed a simple model with globally aggregated stocks of: atmospheric carbon in the form of carbon dioxide, $c_a$; terrestrial carbon, including vegetation and soil carbon, $c_t$; and dissolved inorganic carbon (DIC) in the ocean mixed layer, $c_m$. The model's fourth state variable is global mean surface temperature relative to pre-industrial, $\Delta T = T - T_0$. Compared to Anderies et al. (2013), our model includes more realistic representation of terrestrial and ocean processes but without increase in model complexity, as well as time lags for climate response to $CO_2$.

We now describe the dynamics of the land carbon stock, the ocean carbon stock, and atmospheric carbon and temperature in our model.

## 2.1 Land

Net primary production (NPP) is the net uptake of carbon from the atmosphere by plants through photosynthesis. NPP is expected to increase with concentration of atmospheric carbon dioxide $c_a$. A simple parameterisation of this so-called fertilisation effect is 'Keeling's formula' for global NPP (Bacastow et al., 1973; Alexandrov et al., 2003):

$$\text{NPP}(c_a) = \text{NPP}_0 \left( 1 + K_C \log \frac{c_a}{c_{a0}} \right) \tag{1}$$

Throughout this article, the subscript '0' denotes the value of the quantity at a pre-industrial equilibrium, and 'log' denotes natural logarithm. Keeling's formula incorporates all climate change-related effects on global NPP occurring simultaneously with carbon dioxide changes, for example, precipitation and temperature effects, in addition to fertilisation effects. The curvature of the log function represents limitations to NPP such as changing carbon-use efficiency (Körner, 2003) or nutrient limitations (Zaehle et al., 2010).

At the same time, carbon loss from the world's soils through respiration, $R$, is expected to increase at higher global mean surface temperature, $\Delta T$. We approximate the net temperature response of global soil respiration using the Q10 formalism $R(\Delta T) = R_0 Q_R^{\Delta T/10} c_t/c_{t0}$ (Xu and Shang, 2016), where $Q_R$ is the proportional increase in respiration for a 10 K temperature increase. We assume that pre-industrial soil respiration is balanced by pre-industrial net primary productivity, $R_0 = \text{NPP}_0$. To avoid introducing multiple pools of carbon into the model, we also have to assume that global soil respiration is proportional to total land carbon (rather than soil carbon). Respiration in our model implicitly also includes other carbon-emitting processes such as wildfires or insect disturbances.

It follows that the change in global terrestrial carbon storage is

$$\frac{dc_t}{dt} = \text{NPP}_0 \left( 1 + K_C \log \frac{c_a}{c_{a0}} \right) - \frac{\text{NPP}_0}{c_{t0}} Q_R^{T/10} c_t - \text{LUC}(t).$$

In this expression we have also included loss of terrestrial carbon due to land use emissions $\text{LUC}(t)$. We rearrange this expression to give

$$\frac{dc_t}{dt} = \frac{\text{NPP}_0}{c_{t0}} Q_R^{\Delta T/10} [K(c_a, \Delta T) - c_t] - \text{LUC}(t) \tag{2}$$

where the terrestrial carbon carrying capacity is

$$K(c_a, \Delta T) = \frac{1 + K_C \log \frac{c_a}{c_{a0}}}{Q_R^{\Delta T/10}} c_{t0}. \tag{3}$$





For model simplicity, we do not explicitly model factors affecting terrestrial carbon uptake such as seasonality, species interactions, species functionality, migration, and regional variability.

## 2.2 Ocean

In the upper ocean mixed layer, mixing processes allow exchange of carbon dioxide with the atmosphere. The solubility and

biological pumps then transport carbon from the mixed layer into the deep ocean. Since the residence time of deep ocean carbon is several centuries, we explicitly only model the dynamics of upper ocean carbon while the deep ocean is treated merely as an extremely large carbon reservoir. We include the effects of ocean carbon chemistry, the solubility and biological pumps, and ocean-atmosphere diffusion on upper ocean mixed layer carbon.

Ocean uptake of carbon dioxide from the atmosphere is chemically buffered by other species of dissolved inorganic carbon

such as $HCO_3^-$ and $CO_3^{2-}$, which are produced when dissolved $CO_2$ reacts with water. The reaction of $CO_2$ with water, producing these other species, reduces the partial pressure of $CO_2$ in water allowing for more ocean $CO_2$ uptake before equilibrium with the atmosphere is achieved. The Revelle factor, $r$, is defined as the the ratio of the proportional change in carbon dioxide content to the proportional change in total dissolved inorganic carbon (Sabine et al., 2004; Goodwin et al., 2007). For simplicity, we assume a constant Revelle factor, except for the temperature dependence, $D_T$, of the solubility of

$CO_2$ in sea water. Therefore $CO_2$ diffuses between the atmosphere and ocean mixed layer at a rate proportional to

$$c_a - p(c_m, \Delta T), \tag{4}$$

where

$$p(c_m, \Delta T) = c_{a0} \left( \frac{c_m}{c_{m0}} \right)^r \frac{1}{1 - D_T \Delta T}, \tag{5}$$

since at pre-industrial equilibrium $p(c_{m0}, 0) = c_{a0}$.

There are two main mechanisms by which carbon is transported out of the upper ocean mixed layer into the deep ocean stocks: the solubility and biological pumps. In the solubility pump, overturning circulations exchange mixed layer and deep ocean water. We assume that the large size of the deep ocean means its carbon concentrations are negligibly changed over the 100-year time scales relevant for the model. The net transport of carbon to the lower ocean by the solubility pump can therefore be represented by

$$w_0(1 - w_T \Delta T)(c_m - c_{m0}),$$

where $w_0$ is the (proportional) rate at which mixed layer ocean water is exchanged with the deep ocean and $w_T$ parameterises weakening of the overturning circulation that is expected to occur with future climate change (Collins et al., 2013).

The biological pump refers to the sinking of biomass and organic carbon produced in the upper ocean to deeper ocean layers (Volk and Hoffert, 1985). In the models on which the IPCC reports are based, a weakening of the biological pump is predicted

under climate change, mostly due to a decrease in primary production, in turn due to increases in thermal stratification of ocean





waters (Bopp et al., 2013). We represent this climate-induced weakening in a single approximately linear factor, so that the rate of carbon transported out of the upper ocean mixed layer by the biological pump to lower deep sea layers is given by

$$B(\Delta T) = B_0(1 - B_T \Delta T).$$

As on land, we assume a pre-industrial equilibrium where the biological pump was balanced by transport of carbon back to the

mixed layer by ocean circulation. We neglect deposition of organic carbon to the sea floor and the long time-scale variations in the biological pump that may have contributed to glacial-interglacial cycles (Sigman and Boyle, 2000). We therefore add an additional term $B(\Delta T) - B(0)$ to the transport of carbon from the ocean mixed layer to the deep ocean. Organic carbon that does not sink to the deep ocean is rapidly respired back to forms of inorganic carbon; the ocean mixed layer stock of organic carbon is therefore small, around 3 PgC (Ciais et al., 2013), and we do not count it in the model's carbon balance.

By combining the expressions for the solubility and biological pumps with ocean-atmosphere carbon dioxide diffusion, we obtain the rate of change of ocean mixed layer DIC, $c_m$:

$$\frac{dc_m}{dt} = \frac{Dc_{m0}}{rp(c_{m0}, 0)}(c_a - p(c_m, \Delta T)) - w_0(1 - w_T \Delta T)(c_m - c_{m0}) - B(\Delta T) + B(0), \qquad (6)$$

The coefficient of the first term was chosen such that $1/D$ is the time scale on which carbon dioxide diffuses between the atmosphere and the ocean mixed layer (that is, derivative of the first term with respect to $c_m$, evaluated at the pre-industrial

equilibrium, is $D$).

The carbon content of the deep ocean does not explicitly enter Eq. (6). To evaluate ocean carbon feedbacks, however, we require the change in total ocean carbon content $c_M$ compared to pre-industrial conditions. We calculate this as ocean mixed layer carbon plus carbon transported to the deep ocean by the solubility and biological pumps:

$$\Delta c_M = \Delta c_m + \int^t [w_0(1 - w_T \Delta T)(c_m(t) - c_{m0}) + B(\Delta T) - B(0)] \, dt \qquad (7)$$

We do not explicitly model factors such as the thickness of ocean stratification layers, spatial variation of stratification, nutrient limitations to NPP, or changes in ocean circulation due to wind forcing, freshwater forcing or sea-ice processes (Bernardello et al., 2014).

### 2.3 Atmosphere

Carbon is conserved within the ocean mixed layer, atmospheric and terrestrial carbon stocks. The only processes that affect the

total carbon in the model are human emissions of fossil carbon into the atmosphere and export of carbon into the deep ocean by the solubility and biological pumps,

$$c_a + c_t + c_m = c_{a0} + c_{t0} + c_{m0} + E(t) - \int^t w_0(1 - w_T \Delta T)(c_m - c_{m0}) \, dt - \int^t (B(\Delta T) - B(0)) \, dt. \qquad (8)$$

Increasing atmospheric carbon dioxide levels $c_a$ cause an change in global mean surface temperature, $\Delta T$, compared to its pre-industrial level. To model the response of $\Delta T$, we follow the formulation of Kellie-Smith and Cox (2011), which includes





a logarithmic response as per the Arrhenius law and a delay of time scale $\tau$. Physically, this time delay is primarily due to the heat capacity of the ocean.

$$\frac{dT}{dt} = \frac{1}{\tau}\left(\frac{\lambda}{\log 2}\log\left(\frac{c_a}{c_{a0}}\right) - \Delta T\right). \tag{9}$$

The climate sensitivity $\lambda$ specifies the increase of temperature in response to a doubling of atmospheric carbon dioxide levels.

The climate sensitivity accounts for energy balance feedbacks such as from clouds and albedo. We use the transient climate sensitivity (Collins et al., 2013) as this specifies the response of the climate system over an approximately 100-year time scale (see section 3).



**Table 1.** Model parameters.

| Name | Symbol | Value | Reference/Notes |
|---|---|---|---|
| Pre-industrial atmospheric carbon | $c_{a0}$ | 589 PgC | Ciais et al. (2013) |
| Pre-industrial soil and vegetation carbon | $c_{t0}$ | 1875 PgC | 1325 PgC of soil organic carbon in top metre of soil (Köchy et al., 2015) plus midrange of vegetation carbon estimate by the Ciais et al. (2013). |
| Pre-industrial ocean mixed layer carbon | $c_{m0}$ | 900 PgC | Ciais et al. (2013) |
| Climate sensitivity | $\lambda$ | 1.8 K | Multi-model mean transient climate response (Flato et al., 2013) |
| Climate lag | $\tau$ | 4 yr | Calculations on ocean heat uptake, the primary cause of climate lag, indicate a response time ($e$-folding time) of 4 yr for time scales up to centuries, before deep ocean heat uptake dominates at millennial time scales (Gregory et al., 2015). This result is consistent with simulations that indicate that maximum warming after a $CO_2$ pulse is reached after only a decade (Ricke and Caldeira, 2014) and with results from impulse response model experiments (Joos et al., 2013). |
| Atmosphere-ocean mixed layer $CO_2$ equilibration rate | $D$ | 1 yr$^{-1}$ | Time scale of approximately 1 year, although highly spatially dependent (Jones et al., 2014). |
| Revelle (buffer) factor | $r$ | 12.5 | Williams et al. (2016) |
| Solubility temperature effect | $D_T$ | 4.23%/K | Takahashi et al. (1993); Ciais et al. (2013, p498) |
| Pre-industrial biological pump | $B_0$ | 13 PgC/yr | Ciais et al. (2013) |
| Temperature dependence of biological pump | $B_T$ | 3.2%/K | 12% decrease (Bopp et al., 2013, Fig 9b) after approximately 3.7 K climate change (Collins et al., 2013) |
| Solubility pump rate | $w_0$ | 0.1 yr$^{-1}$ | DIC flux rate from ocean mixed layer divided by DIC stock in mixed layer (Ciais et al., 2013) |
| Weakening of overturning circulation with climate change | $w_T$ | 10%/K | Approximate fit to values reported by Collins et al. (2013, p1095) |
| Terrestrial respiration temperature dependence | $Q_R$ | 1.72 | Raich et al. (2002); Xu and Shang (2016). Based on soil respiration, which contributes the majority of terrestrial ecosystem respiration. |
| Pre-industrial NPP | NPP$_0$ | 55 PgC/yr | Wieder et al. (2015); Sitch et al. (2015) |
| Fertilisation effect | $K_C$ | 0.3 | Estimated by substituting recent NPP $\approx$ 60 PgC/yr (Wieder et al., 2015; Sitch et al., 2015) and recent terrestrial carbon stocks, $c_t \approx c_{t0} + 240$ (Ciais et al., 2013), into Eq. (1). Alexandrov et al. (2003) found that values between 0.3 and 0.4 are compatible with results from a process-based global NPP model. |





**Table 2.** Model validation. Historical changes are carbon stocks in 2011 relative to stocks in 1750 (Ciais et al., 2013) and temperatures in 2012 relative to temperatures in 1880 (Hartmann et al., 2013). Predicted future changes are carbon stocks in 2100 compared to 2012 (Collins et al., 2013) and global mean surface temperatures (GMST) averaged over 2081–2100 relative to 1986–2005 (Collins et al., 2013), under the range of RCP scenarios.

|  | Ocean carbon changes (PgC) | | Land carbon changes (PgC) | | GMST change, $\Delta T$ (K) | |
|---|---|---|---|---|---|---|
|  | IPCC AR5 | Model result | IPCC AR5 | Model result | IPCC AR5 | Model result |
| Historical | $155 \pm 30$ | 95 | $-30 \pm 45$ | 26 | 0.85 [0.65 to 1.06] | 0.82 |
| RCP2.6 | 150 [105 to 185] | 174 | 65 [-50 to 195] | 67 | 1.0 [0.3 to 1.7] | 0.5 |
| RCP4.5 | 250 [185 to 400] | 243 | 230 [55 to 450] | 135 | 1.8 [1.1 to 2.6] | 1.2 |
| RCP6 | 295 [265 to 335] | 278 | 200 [-80 to 370] | 168 | 2.2 [1.4 to 3.1] | 1.7 |
| RCP8.5 | 400 [320 to 635] | 340 | 180 [-165 to 500] | 207 | 3.7 [2.6 to 4.8] | 2.4 |

## 3 Model parameterisation and validation

Our climate-carbon cycle model has twelve parameters, four state variables and three nontrivial initial conditions (by definition, the initial value of $\Delta T$ is 0). Parameters for the response of climate to carbon dioxide ($\lambda, \tau$) and two parameters of the response of the ocean to changing temperature ($B_T$ and $w_T$) were set based on the output of atmosphere-ocean global circulation models. All other parameters are based on historical observations of the global carbon cycle (Table 1).

Unless otherwise noted, we perform emissions-based model runs using harmonized historical data and future RCP scenarios on fossil fuel emissions [$E(t)$] and land use emissions [LUC($t$)] (Meinshausen et al., 2011b). While the focus of our study is on future climate change, from the present day until 2100, we begin simulations in 1750 to compare our model against historical observations. Time series of the model output are displayed in Fig. 2. Model solutions were approximated in continuous time.



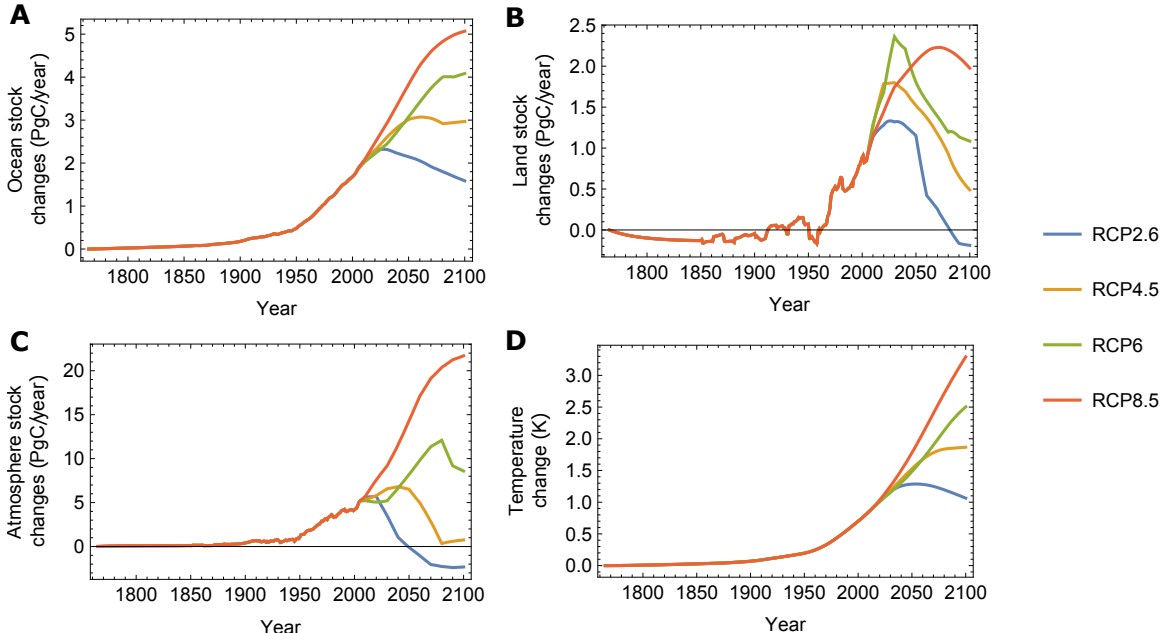

**Figure 2.** Model output under forcing from different RCP scenarios: (a) land carbon stock change, (b) ocean carbon stock changes, (c) atmospheric carbon stock change, and (cd global mean surface temperature change.

## 4 Feedback analysis

Our climate-carbon cycle model is sufficiently simple that the strengths of its feedbacks can be estimated analytically. Such computations are useful since the resulting symbolic expressions can be used to identify how parameters of interest affect feedback strengths and model dynamics. In this section we introduce definitions of feedback strengths, calculate climate-carbon cycle feedbacks analytically and numerically, and estimate feedback nonlinearities.

### 4.1 Definitions

There are multiple measures of carbon cycle feedbacks currently in use. We here review three of the most common measures.

Consider an emission of $E$ PgC over some time period to the atmosphere. In the absence of carbon cycle feedbacks, the atmospheric carbon content would increase by $\Delta c_a^{\mathrm{off}} \equiv E$. With a feedback switched on, the atmospheric carbon content would actually change by $\Delta c_a^{\mathrm{on}}$. The *feedback factor* is (Zickfeld et al., 2011)

$$F = \frac{\Delta c_a^{\mathrm{on}}}{\Delta c_a^{\mathrm{off}}}. \tag{10}$$

Out of the total atmospheric carbon change $\Delta c_a^{\mathrm{on}}$, the carbon cycle feedback contributes (Hansen et al., 1984)

$$\Delta c_a^{\mathrm{feedback}} = \Delta c_a^{\mathrm{on}} - \Delta c_a^{\mathrm{off}}. \tag{11}$$



Gain is the change in a feedback to atmospheric carbon content caused by changes in atmospheric carbon content:

$$g = \frac{\Delta c_a^{\text{feedback}}}{\Delta c_a^{\text{on}}}. \tag{12}$$

Gain and feedback factor are related by

$$F = \frac{1}{1-g}. \tag{13}$$

5       An alternative formalism, introduced by Friedlingstein et al. (2006), allows feedbacks to be characterised from carbon cycle model output. Climate models are not required, except as a forcing to the carbon cycle model. The formalism relates the changes in terrestrial and marine carbon stocks to changes in global mean temperature and atmospheric carbon dioxide as follows:

$$\Delta c_t = \beta_L \Delta c_a + \gamma_L \Delta T \tag{14}$$

$$\Delta c_M = \beta_O \Delta c_a + \gamma_O \Delta T. \tag{15}$$

Here the $\beta_L$ and $\beta_O$ feedback parameters are the land and ocean, respectively, carbon sensitivities to atmospheric carbon dioxide changes $\Delta c_a$. Likewise, $\gamma_L$ and $\gamma_O$ are the land and ocean, respectively, carbon sensitivities to temperature changes $\Delta T$. Note that $c_M$ denotes the total marine carbon stock, both mixed layer and deep ocean. The differences $\Delta c_a$, etc., are usually calculated over the duration of a simulation. To isolate the $\beta$ and $\gamma$ feedback parameters, simulations are conducted

with biogeochemical coupling only and with radiative coupling only (Gregory et al., 2009).

      In both the formalisms introduced thus far, the feedback measures are calculated by examining the changes in carbon stocks at the end point of model simulations. In contrast, Boer and Arora (2009) estimate sensitivities $\Gamma$ and $B$ of the instantaneous carbon *fluxes* from atmosphere to land and ocean:

$$\frac{dc_t}{dt} = B_L \Delta c_a + \Gamma_L \Delta T \tag{16}$$

$$\frac{dc_M}{dt} = B_O \Delta c_a + \Gamma_O \Delta T. \tag{17}$$

These feedback parameters $B$ and $\Gamma$ are usually computed for all time points during a simulation, again using biogeochemically coupled and radiatively coupled simulations.

      The two sets of parameters $(B,\Gamma)$ and $(\beta,\gamma)$ are related by

$$\beta \Delta c_a = \int B \Delta c_a dt \tag{18}$$

$$\gamma \Delta T = \int \Gamma \Delta T dt. \tag{19}$$

Accordingly, Boer and Arora (2013) refer to $B$ and $\Gamma$ as *direct feedback parameters* and to $\beta$ and $\gamma$ as *time-integrated feedback parameters*.




## 4.2 Analytical feedback strengths based on equilibrium changes

Analytical approximations to the strengths of carbon cycle feedbacks in our model require choosing a time scale on which the feedbacks will be calculated. Numerically estimated feedback factors [Eq. (10)] and time-integrated feedback parameters [Eqs. (14-15)] are conventionally calculated using carbon stock changes over 100 years or more. Responses on the longest

time scales of our model are therefore most relevant if our analytical approximates are to approximate numerically calculated values. While recognising that the Earth's climate system is presently far from equilibrium, we use changes in the equilibrium state of the model to approximate model responses over long time scales.

We analytically calculate the gains associated with each of the feedback loops in Fig. 1 as follows. We calculate the sensitivity (mathematically, partial derivative) of the equilibrium value of each quantity in the feedback loop with respect to the preceding

quantity in the loop. We form the product of the derivatives (as per the chain rule of differentiation) to estimate the gain of that feedback loop. For example, to calculate the land climate-carbon gain we calculate the sensitivity of equilibrium temperature with respect to changes in atmospheric carbon content ($\partial T^*/\partial c_a$), multiplied by the sensitivity of equilibrium terrestrial carbon with respect to changes in temperature ($\partial c_t^*/\partial T$), multiplied by the sensitivity of equilibrium atmospheric carbon with respect to changes in terrestrial carbon ($\partial c_a^*/\partial c_t$).

$$\text{Land climate-carbon equilibrium gain} \quad g_{TL}^* \equiv \frac{\partial T^*}{\partial c_a} \frac{\partial c_t^*}{\partial T} \frac{\partial c_a^*}{\partial c_t}$$

$$\text{Land concentration-carbon equilibrium gain} \quad g_L^* \equiv \frac{\partial c_t^*}{\partial c_a} \frac{\partial c_a^*}{\partial c_t}$$

$$\text{Ocean climate-carbon equilibrium gain} \quad g_{TO}^* \equiv \frac{\partial T^*}{\partial c_a} \frac{\partial c_m^*}{\partial T} \frac{\partial c_M}{\partial c_m} \frac{\partial c_a^*}{\partial c_M}$$

$$\text{Ocean concentration-carbon equilibrium gain} \quad g_O^* \equiv \frac{\partial c_m^*}{\partial c_a} \frac{\partial c_M}{\partial c_m} \frac{\partial c_a^*}{\partial c_M}$$

The subscript $T$ denotes that the feedback involves temperature. Asterisks (*) denote equilibrium quantities. From these gains,

the feedback factors $F_{TL}^*$, $F_L^*$, $F_{TO}^*$ and $F_O^*$ can be calculated using Eq. (13). We label these gains and feedbacks factors $g^*$ and $F^*$, respectively, to denote they are based on an equilibrium approximation, not directly from transient simulations as estimated by Zickfeld et al. (2011). We calculate $\frac{\partial c_m^*}{\partial c_a} \frac{\partial c_M}{\partial c_m}$ rather than simply $\frac{\partial c_M^*}{\partial c_a}$ as it is $c_m$ that is a state variable in our model, from which we then estimate $c_M$.

The derivatives of $c_a^*$ are trivial to calculate: by carbon balance, $\frac{\partial c_a^*}{\partial c_t} = \frac{\partial c_a^*}{\partial c_M} = -1$. To calculate $\frac{\partial c_m^*}{\partial T}$ we set $0 = \frac{dc_m}{dt} \equiv$

$f(c_m^*, c_a, T)$, use the chain rule to obtain $0 = \frac{\partial f}{\partial c_m^*} \frac{\partial c_m^*}{\partial T} + \frac{\partial f}{\partial T}$, and then solve for $\frac{\partial c_m^*}{\partial T}$ in terms of the partial derivatives of $f$. Similar procedures provide $\frac{\partial c_m^*}{\partial c_a}$ and the derivatives of $T^*$ and $c_t^*$.

The remaining derivative is $\partial c_M/\partial c_m$. Carbon sunk into the deep ocean is substantial and cannot be neglected. Deep ocean carbon storage equilibrates on time scales of millennia or more, however, far longer than the time scales of interest in this model (we therefore write $\partial c_M/\partial c_m$ not $\partial c_M^*/\partial c_m$). We therefore cannot use the same equilibrium approach as for the other

variables. Instead, we use Eq. (7) with the following approximations. First, we neglect the temperature dependence of the biological pump and the rate of the overturning circulation, as for this derivative we are primarily interested in the effects of changing carbon stocks, not temperatures. Second, let us assume a scenario where the trajectory of ocean mixed layer DIC



$c_m$ can be approximated by a linear increase from $c_{m0}$ to $c_m$ over a time interval $t_{\text{lin}}$. We estimate this time interval by $t_{\text{lin}} = (c_m(t_{\text{end}}) - c_{m0})/c'_m(t_{\text{end}})$ using the value $c_m$ and gradient $c'_m$ at the end of the simulation period. Using this approximation and Eq. (7),

$$\frac{\partial c_M}{\partial c_m} \approx 1 + M \tag{20}$$

where $M = w_0 t_{\text{lin}}/2$. The value of $M$ will be strongly scenario-dependent.

We analytically estimate equilibrium versions of the time-integrated feedback parameters of Friedlingstein et al. (2006) using a similar approach:

$$\gamma_L^* = \frac{\partial c_t^*}{\partial T}$$
$$\beta_L^* = \frac{\partial c_t^*}{\partial c_a}$$
$$\gamma_O^* = \frac{\partial c_m^*}{\partial T} \frac{\partial c_M}{\partial c_m}$$
$$\beta_O^* = \frac{\partial c_m^*}{\partial c_a} \frac{\partial c_M}{\partial c_m}.$$

Since the ocean component of the model has multiple processes that respond to temperature, some analytical forms were too complicated for easy visual inspection (Table A1). We derived approximate analytical feedbacks by comparing the magnitudes of terms in the numerator and denominator of the feedback measures, and by expanding numerators in power series of $D_T T$

and $c_a/c_{a0}$.

### 4.3    Analytical feedback strengths based on carbon fluxes

We estimate the direct feedback parameters as follows:

$$\Gamma_L^* = \left.\frac{dc_t}{dt}\right|_{c_a=c_{a0}} \frac{1}{\Delta T}$$
$$B_L^* = \left.\frac{dc_t}{dt}\right|_{\Delta T=0} \frac{1}{c_a - c_{a0}}$$
$$\Gamma_O^* = \left.\frac{dc_M}{dt}\right|_{c_a=c_{a0}} \frac{1}{\Delta T}$$
$$B_O^* = \left.\frac{dc_M}{dt}\right|_{\Delta T=0} \frac{1}{c_a - c_{a0}}.$$

Here $dc_t/dt$ and $dc_M/dt$ denote the atmosphere-land and atmosphere-ocean fluxes. The subscript $\Delta T = 0$ denotes a biogeo-chemically coupled (and radiatively decoupled) simulation and $c_a = c_{a0}$ denotes a radiatively coupled (and biogeochemically decoupled) simulation. We use the

The values of the feedback parameters are strongly scenario-dependent (Arora et al., 2013). To calculate the direct feedback parameters, we assume a standard $CO_2$-quadrupling concentration pathway in order to compare our results with Arora et al. (2013). This scenario has $c_a(t) = c_{a0}a^t$ where $a = 1.01$. In this scenario, $\frac{1}{c_a}\frac{dc_a}{dt} = \log a$ and, ignoring an initial exponential transient, $\frac{dT}{dt} = \lambda \log a/\log 2$.





For the atmosphere-land carbon flux, the calculation is straightforward under the following assumptions. We assume that $NPP_0/c_{t0} \gg \log a$ so that $c_t$ tracks its carrying capacity $c_t \approx K$ [Eq. (2)]. We also ignore land use change, so that $\frac{dc_t}{dt} \approx \frac{dK}{dt}$. Then we calculate $\frac{dK}{dt}|_{c_a=c_{a0}} = \frac{\partial K}{\partial T}\frac{dT}{dt}$ and $\frac{dK}{dt}|_{\Delta T=0} = \frac{\partial K}{\partial c_a}\frac{dc_a}{dt}$.

While the atmosphere-ocean flux could be read off directly from the first term of Eq. (6), this form is however not particularly useful. As it involves a small difference between two large quantities, $c_a$ and $p(c_m, \Delta T)$, the size of the difference can only be estimated from numerical results and gives no immediate insight into how it depends on parameters. Furthermore, we seek to compare our analytical results to the results presented by Arora et al. (2013), in which the feedback parameters are presented as functions of $c_a$ or $\Delta T$ only (not $c_m$).

We instead derive an approximation based on time scale separation as follows. The characteristic time scales of atmosphere-ocean diffusion, solubility pump, biological pump and human emissions into the atmosphere are $D$, $w_0$, $B_0/c_{m0}$ and $\log a$ respectively. These rates are ordered $D \gg w_0 \gg \log a, B_0/c_{m0}$. Since atmosphere-ocean diffusion is the fastest process, ocean mixed layer carbon content rapidly gains an equilibrium $c_m = p^{-1}(c_a, \Delta T)$ with respect to atmospheric carbon content, where $p^{-1}(c_a, \Delta T)$ is the solution to $c_a = p(c_m, \Delta T)$. Ocean and atmosphere partial pressures are kept out of equilibrium by the next fastest process: the solubility pump. On the time scale of our model, the atmosphere-ocean flux is therefore controlled by the solubility pump, with diffusion providing a rapid coupling between ocean mixed layer and atmosphere. An approximation for the atmosphere-ocean flux is therefore $dc_M/dt \approx w_0(1 - w_T \Delta T)(p^{-1}(c_a, \Delta T) - c_{m0})$, which upon substitution into the definitions of $B_O^*$ and $\Gamma_O^*$ gives the forms in Table 3.

## 4.4 Numerical estimation of feedback strengths

In addition to the analytical approximations to carbon cycle feedbacks derived from our model, we also estimate feedback factors from direct numerical simulations of our model. To compare the results of our model to previous studies, we use the following scenarios. To compare our results with the time-integrated feedback parameters reported by Friedlingstein et al. (2006) and the feedback factors and gains of Zickfeld et al. (2011), we employ the SRES A2 emissions scenario used in those articles. To compare our results with the direct feedback parameters of Arora et al. (2013), we use the doubling $CO_2$ concentration scenario used in that article. For each scenario, we perform three simulations:

1. Fully coupled simulation.

2. Completely uncoupled simulation, giving $c_a^{off}(t) = c_{a0} + \int^t E(t)dt$ for the emissions-driven scenario and the specified concentration pathway for concentration-driven scenario.

3. Biogeochemically coupled simulation. We switch off feedbacks involving temperature response to atmospheric $CO_2$, by setting $\lambda = 0$. From this simulation we estimate the carbon-concentration feedback factors via land $F_L = \Delta c_a^{on}/\Delta c_a^{off} = 1 - \Delta c_t/\Delta c_a^{off}$ and ocean $F_O = \Delta c_a^{on}/\Delta c_a^{off} = 1 - \Delta c_M/\Delta c_a^{off}$, time-integrated feedback parameters $\beta_L = \Delta c_t/\Delta c_a$ and $\beta_O = \Delta c_M/\Delta c_a$, and direct feedback parameters $B_L(t) = \frac{dc_t}{dt}/(c_a - c_{a0})$ and $B_O(t) = \frac{dc_M}{dt}/(c_a - c_{a0})$.





**Table 3.** Feedback analysis. Gains ($g$), feedback factors ($F$), time-integrated feedback parameters ($\gamma$ and $\beta$) and direct feedback parameters ($\Gamma$ and $B$) were calculated analytically and numerically. Analytical ocean feedbacks are approximations of the exact forms in Table A1 (see Sec. 4.2. Exact forms were used to calculate numerical values. In this table, $p \equiv p(c_m, T)$. Units of the climate-carbon integrated feedback parameters are PgC/K and concentration-carbon integrated feedback parameters are PgC/ppm. Ranges for analytical results are written in the form (value at start of simulation) to (value at end of simulation). Emissions scenarios are as indicated; land use emissions were assumed to be zero.

| Feedback measure | Land climate-carbon feedback | Ocean climate-carbon feedback | Land conc.-carbon feedback | Ocean conc.-carbon feedback |
|---|---|---|---|---|
| Gain, analytical expression | $\dfrac{\lambda c_{t0}\left(1 + K_C \log \frac{c_a}{c_{a0}}\right)\log Q_R}{10 c_a Q_R^{\Delta T/10} \log 2}$ | $\dfrac{\lambda D D_T (1+M) c_m}{r c_a (1 - D_T \Delta T)\log 2}$ | $-\dfrac{c_{t0} K_C}{c_a Q_R^{\Delta T/10}}$ | $-\dfrac{(1+M)c_m}{rp}$ |
| Feedback factor (numerical scenario: SRES A2) ($> 1$ amplifies climate change; $< 1$ dampens climate change) | | | | |
| - estimate from analytical gain | 1.81 to 1.18 | 1.07 to 1.03 | 0.51 to 0.81 | 0.59 to 0.80 |
| - from simulation | 1.15 | 1.10 | 0.74 | 0.67 |
| - Zickfeld et al. (2011) | 1.25 | 1.22 | 0.66 | 0.71 |
| Time-integrated feedback parameter (numerical scenario: SRES A2) ($< 0$ amplifies climate change; $> 0$ dampens climate change) | | | | |
| - analytical expression | $-\dfrac{c_{t0}\log Q_R}{10 Q_R^{\Delta T/10}}$ | $-\dfrac{D(1+M)c_m D_T}{r(1 - D_T \Delta T)}$ | $\dfrac{c_{t0} K_C}{c_a}$ | $\dfrac{(1+M)c_m}{rp}$ |
| - estimate from analytical form | -102 to -86 | -15 to -21 | 2.04 to 0.60 | 1.48 to 0.51 |
| - from simulation | -74 | -48 | 0.73 | 1.04 |
| - Zickfeld et al. (2011) | -129 | -32 | 1.12 | 0.86 |
| - Friedlingstein et al. (2006) | -79 (-20 to -177) | -30 (-14 to -67) | 1.35 (0.2 to 2.8) | 1.13 (0.8 to 1.6) |
| Direct feedback parameter (numerical scenario: $CO_2$ doubling) ($< 0$ amplifies climate change; $> 0$ dampens climate change) | | | | |
| - analytical expression | $-\dfrac{c_{t0}\lambda \log Q_R \log a}{10 \Delta T \log 2}$ | $-\dfrac{w_0 c_{m0} D_T}{r}$ | $\dfrac{c_{t0} K_C \log a}{c_a - c_{a0}}$ | $\dfrac{w_0 c_{m0} c_a}{r c_{a0}(c_a - c_{a0})}$ |
| - estimate from analytical form | Fig. A1a | Fig. A1b | Fig. A1a | Fig. A1b |
| - from simulation | Fig. A1a | Fig. A1b | Fig. A1a | Fig. A1b |
| - Arora et al. (2013) | | see text | | |
| Nonlinearity | -0.43 | -0.06 | 0.03 | 0.02 |

4. Radiatively coupled simulation. We switch off feedbacks involving the carbon cycle, by setting $K_C = 0$ and changing the $c_a$ in Eq. (6) to $c_{a0}$. From this simulation we estimate the carbon-climate feedback factors $F_{TL} = 1 - \Delta c_t / \Delta c_a^{\text{off}}$ and $F_{TO} = 1 - \Delta c_M / \Delta c_a^{\text{off}}$, time-integrated feedback parameters $\gamma_L = \Delta c_t / \Delta T$ and $\gamma_O = \Delta c_M / \Delta T$, and direct feedback parameters $\Gamma_L(t) = \frac{dc_t}{dt} / \Delta T$ and $\Gamma_O(t) = \frac{dc_M}{dt} / \Delta T$.

5 **4.5 Feedback nonlinearity**

Zickfeld et al. (2011) found, in emissions-driven scenarios, that the fully coupled simulation sunk more terrestrial and marine carbon than the sum of the biogeochemically and radiatively coupled scenarios. They refer to this difference as the non-linearity



of the feedback, with the land sink contributing 80% of the nonlinearity and the ocean sink 20%. Our analytical expressions for the feedbacks can be used to obtain an alternative measure of feedback nonlinearity.

We evaluate the nonlinearity in the ocean and land climate-carbon feedbacks by $F^*_{TO}(c_a, c_m, c_t, \Delta T) - F^*_{TO}(c_{a0}, c_{m0}, c_{t0}, \Delta T)$ and $F^*_{TL}(c_a, c_m, c_t, \Delta T) - F^*_{TL}(c_{a0}, c_{m0}, c_{t0}, \Delta T)$, respectively, where the $F^*(c_{a0}, c_{m0}, c_{t0}, \Delta T)$ are analytical approxima-

tions of feedback factors from a radiatively coupled simulation (all carbon stocks are fixed at pre-industrial levels). We evaluate the nonlinearities in the ocean and land concentration-carbon feedbacks by $F^*_O(c_a, c_m, c_t, \Delta T) - F^*_O(c_a, c_m, c_t, 0)$ and $F^*_L(c_a, c_m, c_t, \Delta T) - F^*_L(c_a, c_m, c_t, 0)$, respectively, where the $F^*(c_a, c_m, c_t, 0)$ are analytical approximations of feedback factors from a biogeochemically coupled simulation (temperature is fixed at its pre-industrial level). These expressions indicate the effect of temperature and atmospheric carbon on the concentration-carbon and climate-carbon feedbacks, respectively, We

used the SRES A2 scenario.

## 5 Results and Discussion

### 5.1 Model evaluation

Most predictions of our model are within the range of model predictions produced for the IPCC's Fifth Assessment Report (Table 2). Our model estimates around 55 PgC more historical land carbon uptake than the IPCC multi-model mean, possibly

due to our simplification to a single land carbon pool. Because it omits radiative forcing due to greenhouse gases other than $CO_2$, our model consistently underestimates future temperature changes, although in all except the RCP8.5 scenario the projections are within the IPCC model range. The purpose of our model is not to precisely predict future climate change, but to serve as an approximate, mechanistically based emulator of the carbon cycle component of Earth System Models (see Sec. 2). We conclude that the model emulates historical observations and future projections of Earth System Models with sufficient

accuracy for this purpose.

### 5.2 Feedback analysis

Both feedback measures calculated directly from our model simulations and measures estimated from analytical approximations match well with output from Earth System Models [Table 3; compare also Fig. A1 with figures 4-5 of Arora et al. (2013) for direct feedback parameters]. This agreement serves as additional validation of our model as well as validation of the

approximations used to calculate analytical feedback factors.

The approximate analytical expressions for the three different feedback measures all have similar dependences on state variables and parameters. All measures of the land climate-carbon feedback have dependence of the form $c_{t0} \log Q_R / Q_R^{\Delta T/10}$. The ocean climate-carbon feedbacks all have the form $w_0 D_T c_{m0} / r$ (since $1 + M \sim w_0$ and ocean mixed layer carbon $c_m \approx c_{m0}$ to within 10 % over the duration of the simulation). The land concentration-carbon feedback has the form $c_{t0} K_C / c_a$ and the

ocean concentration-carbon feedbacks have the form $w_0 c_m / r c_a$ (since $p(c_m, \Delta T) \approx c_a$). We conclude that for each type of carbon cycle feedback, all three feedback formalisms detect similar features of the climate-carbon system.



The analytical expressions provide rapid insight into how feedback strengths depend on state variable and parameter values that could otherwise only be studied numerically or by qualitative reasoning. The analytical forms show that increasing Revelle factor $r$, as is likely to occur in an increasingly acidic ocean (Sabine et al., 2004), will decrease the strengths of ocean climate-carbon and concentration-carbon feedbacks. Weakening overturning circulation, via $w_0$, would also decrease the strength of

the ocean carbon cycle feedbacks. On land, the parameters $Q_R$ and $K_C$ control the terrestrial carbon cycle feedbacks.

We can compare likely trends in feedback strengths from the analytical expressions. In the ocean, the destabilising ocean climate-carbon feedback is almost constant, while the ocean concentration-carbon feedback weakens with $c_m$ (since $c_m/p(c_m, \Delta T) \sim c_m^{1-r}$). Similarly, the destabilising land climate-carbon feedback will weaken less than the stabilising concentration-carbon feedback (under $CO_2$ doubling, $\sim Q_R^{-\Delta T/10}$ weakens by 9% at the new temperature equilibrium while $\sim 1/c_a$ weakens by

50%). This difference between the land climate-carbon and concentration-carbon feedbacks stems from the differing curvatures of $K(c_a, \Delta T)$ as a function of $\Delta T$ (close to linear) and $c_a$ (concave). We conclude that under continued carbon emissions, both land and ocean feedbacks will overall become more positive.

Where multiple processes contribute in parallel to a feedback, inspection of analytical forms can indicate the relative contributions of the different processes to the feedback. In the ocean component of the model, $CO_2$ solubility, the biological pump,

and the solubility pump are all temperature-dependent and therefore contribute to the ocean climate-carbon feedback. Terms in the numerators of the exact forms of $g_{TO}$ and $\gamma_O$ (Table A1) correspond to these three processes. Substituting parameters and typical values for state variables into these three terms show that the temperature dependence of $CO_2$ solubility contributes most to these climate-carbon feedbacks.

### 5.3   Feedback nonlinearity

As shown in Sec. 4.5, our analytical feedback expressions enable a new way of estimating feedback nonlinearities that is not possible from direct numerical simulation. Since the sum of the four nonlinearities is negative (Table 3), we conclude that summing feedbacks found by individual decoupled simulations will overestimate the atmospheric carbon levels, that is, underestimate land and ocean sinks. This result matches the findings of Zickfeld et al. (2011) and Matthews (2007). Terrestrial feedbacks contributed 91% of the total nonlinearity in our model, compared to 80% reported by Zickfeld et al. (2011). Fur-

thermore, we can distinguish the nonlinearities in the climate-carbon and concentration-carbon feedbacks. We found that the nonlinearity in the terrestrial carbon-climate feedback was almost ten times larger than any other (Table 3). By inspecting the analytical derivation of the gains we conclude that this dominance is likely due to a combination of two reasons: First, due to the sensitivity of temperature to carbon dioxide, $\partial T/\partial c_a = \lambda/c_a \log 2$, the carbon-climate feedbacks are much more sensitive to $c_a$ than the concentration-carbon feedbacks are to $\Delta T$. Second, the nonlinearity in the land climate-carbon feedback is

larger than the ocean climate-carbon feedback because its feedback factor is larger and therefore more sensitive to changes in gain (see Eq. (12)). We conclude that care must be taken when summing results for feedbacks from decoupled simulations, especially for simulations involving land feedbacks.



## 6 Conclusions

Earth System Models span a wide variety of complexity. Here, we constructed a highly stylised, globally aggregated climate-carbon cycle model. Despite the model's simplicity—just four state variables—the model emulated globally aggregated historical trends and future projections of Earth System Models. The model's simple form allowed climate-carbon cycle feedbacks

to be estimated analytically, providing mechanistic insight into these processes. For example, we showed that carbon-climate feedbacks are less sensitive than carbon-concentration feedbacks; on land, this is due to the shape of $K(c_a, \Delta T)$. The simple but accurate climate-carbon cycle model could be a starting point for model-based investigations of Earth system processes that are too poorly understood to be incorporated in more complex models.

Stylized models such as ours have significant value in policy contexts. When confronted with difficult policy decisions

involving long time periods and significant uncertainty, collaborative work with scientists allows policy makers to identify and clarify the impacts of various policy actions. In this context, the utility of a model is to increase stakeholders' understanding of a system and its dynamics under various conditions (Voinov and Bousquet, 2010; Anderies, 2005). This is in stark contrast to the use of more complex, detailed models to predict impacts of policies where mechanisms underlying dynamics and trade-offs are not transparent, and quick explorations with stakeholders are not practical. The utility of a stylised model is in facilitating

a learning process rather than in 'accurately' predicting outcomes.

We foresee at least two strands of valuable future research based on the climate-carbon cycle model developed in this paper. First, our climate-carbon cycle model could be extended by including further processes relevant on different time-scales of interest for Earth system analysis. This would enable a more in-depth analytical analysis of the feedback strengths and gains relating to other aspects of Earth system dynamics, such as the Earth's energy balance, albedo changes, the cryosphere,

nutrient cycles and even societal feedbacks. The task of characterizing the Anthropocene as an epoch could thus move beyond qualitative comparison of human-impact trends to an improved characterisation the feedbacks that maintain different Earth system 'regimes'.

Second, the model could comprise a 'workbench' for the systemic understanding of planetary boundary interactions and, hence, generate crucial insights into the structure of the safe operating space for humanity delineated by the planetary bound-

aries (Rockström et al., 2009; Steffen et al., 2015). Such extensions should focus on linking the core abiotic and biotic dimensions of the planetary boundaries framework. The present lack of well-developed model representations of the dynamics and ecosystem structure of biosphere diversity, heterogeneity and resilience, despite ongoing efforts in this direction (Purves et al., 2013; Bartlett et al., 2016; Sakschewski et al., 2016), emphasises the need for a more conceptual understanding of biosphere integrity, its vulnerability to anthropogenic perturbation, and its role for Earth system resilience.

*Author contributions.* SJL, JMA, SEC, JFD, IF, KR, JR and WS designed the research. SJL, JFD, IF, TG and CB constructed the model. SJL analysed the model. All authors wrote the paper.

*Competing interests.* The authors declare that they have no conflict of interest.

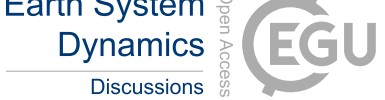



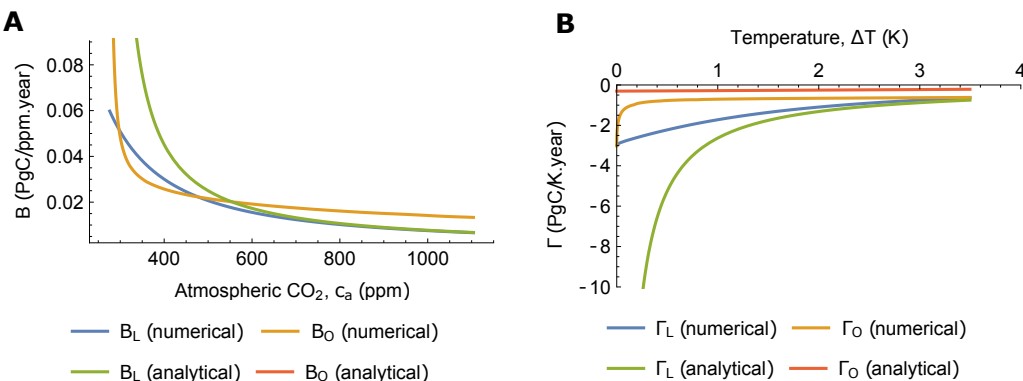

**Figure A1.** Direct feedback parameters, (a) concentration-carbon feedbacks and (b) climate-carbon feedbacks.

**Table A1.** Exact forms for ocean feedbacks.

| Feedback measure | Ocean climate-carbon feedback | Ocean concentration-carbon feedback |
|---|---|---|
| Gain | $\dfrac{\lambda(1+M)}{c_a \log 2} \dfrac{\frac{pc_{m0}DD_T}{c_{a0}r(1-D_T\Delta T)} - B_0 B_T - w_0 w_T(c_m - c_{m0})}{\frac{pDc_{m0}}{c_m c_{a0}} + w(1 - w_T \Delta T)}$ | $-\dfrac{c_{m0}D}{c_{a0}r} \dfrac{1+M}{\frac{pDc_{m0}}{c_m c_{a0}} + w(1 - w_T \Delta T)}$ |
| Time-integrated feedback parameter | $-\dfrac{\frac{pc_{m0}DD_T}{c_{a0}r(1-D_T\Delta T)} - B_0 B_T - w_0 w_T(c_m - c_{m0})}{\frac{pDc_{m0}}{c_m c_{a0}} + w(1 - w_T \Delta T)}$ | $\dfrac{c_{m0}D}{c_{a0}r} \dfrac{1+M}{\frac{pDc_{m0}}{c_m c_{a0}} + w}$ |
| Direct feedback parameter | $\dfrac{w(1 - w_T\Delta T)c_{m0}\left((1 - D_T\Delta T)^{1/r} - 1\right)}{\Delta T}$ | $\dfrac{wc_{m0}\left(\left(\frac{c_a}{c_{a0}}\right)^{\frac{1}{r}} - 1\right)}{c_a - c_{a0}}$ |

*Acknowledgements.* The research leading to these results has received funding from the Stordalen Foundation via the Planetary Boundary Research Network (PB.net), the Earth League's EarthDoc programme, the Leibniz Association (project DOMINOES), European Research Council Synergy project Imbalance-P (grant ERC-2013-SyG-610028), Project Grant 2014-589 from the Swedish Research Council Formas, and a core grant to the Stockholm Resilience Centre by Mistra. We thank Malin Ödalen for her comments on the manuscript.



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
