# Peer review of "Analytically tractable climate-carbon cycle feedbacks under 21st century anthropogenic forcing"

_Earth System Dynamics, 2017_

## Short Comment (SC1) · 12 Nov 2017

I like this model a lot. You may improve its exposision marginally and make it less time-dependent by introducing a fifth state variable, cd = deep ocean carbon stock, and rewrite eq. (8) as two ODEs, one for cd (containing the two integrals) and one for ca. This way the model's remaining time dependency is only on the two "control" variables E(t) and LUC(t) and it may thus be analysed more easily using tools from bifurcation analysis or topology of sustainable management (see this special issue).

---

## Referee Comment (RC1) · C.D. Jones (Referee) · 21 Nov 2017

This is a nicely designed study, and well presented manuscript which attempts to develop and document a simple ("stylized") model of the global climate-carbon cycle system, but in a way which enables analytical analysis of its behaviour and feedback mechanisms. The intended aim is to facilitate improved understanding of the system dynamics and more readily quantify which processes contribute to certain feedbacks and long-term responses.

Overall, I very much like the approach and the intention – such modelling studies can develop improved insight and can strip back confounding and complicating issues of

model complexity to reveal more fundamental underlying behaviour.

I have a few comments below and a few queries about the extent to which the intentions have been realised – I think these can be readily addressed with revised text and some more context/explanation.

Chris Jones.

Overarching comments:

1. There are numerous simple/stylized carbon cycle models in the literature. You cite Raupach (2013) which I like very much. There is also one I developed and have published with several times, including as recently as last year (Jones et al., Tellus, 2003; Jones et al., Tellus, 2006; Jones et al., ERL, 2016). Then there is the MAGICC model often used in IAMs, the Joos IRF, and also the Oscar model which some of the current authors know very well. So I wonder if a bit more explanation is needed for why a new model is required – could you start from one of the existing ones and achieve the same thing? I guess your main driver is the ability to analytically derive the feedback functions – but it's not clear to me the same is not possible from these previous models (I haven't tried it with the Jones et al simple model, but may do!) – Mike Raupach derived eigenmodes, so I would imagine feedback metrics could follow also, but again I haven't tried.

2. On a similar line – I was looking forward to seeing the analytical derivation of feedback factors, but then realised this was not as "tractable" as your title suggests, and you have to make a lot of assumptions in your sections 4.2 and 4.3. This seems a shame – if this is the case, have you not lost your unique and attractive feature? The resulting expressions are still useful, but not as analytical as you suggest. I also wondered, on seeing the expressions in table 3, if you had compared these to the expressions of Ric Williams et al – who have done a similar based assessment of terms controlling ocean heat/carbon uptake and TCRE. (See e.g., Richard G Williams et al 2016 Environ. Res. Lett. 11 015003)

3. You discuss (page 18, line 16) that you might want to develop this model further to include other mechanisms and forcings. Modellers have attempted a two-box approach for the ocean – by splitting upper and deep ocean before. But I would suggest maybe a two-box approach for the land also – not splitting veg vs soil, but maybe by tropics and extra-tropics as these can have very different responses (even by sign) to changes in climate. IPCC AR5 fig 6.22 shows the latitudinal distribution of "gamma" – there is a clear change of sign towards high latitudes, and so a tropics/high-latitude 2-box approach might be a nice (and novel) development

4. In a few places, you discuss non-linearities – this is good, and important to bring out. It's not necessarily true that land dominates the non-linearity, but this does show up in your rapid-forcing transient runs. If you ran longer, or with slower changing forcing, the ocean would have more chance to exhibit these too – see, e.g. Schwinger et al (J. Climate, 2014). Hence both land and ocean can have pronounced non-linearities and for this reason, C4MIP took the decision to move back towards the Friedlingstein definition of feedbacks as the difference between COUPLED and BGC runs (see Jones et al., GMD, 2016 – documentation of C4MIP). Hence this differs from the Arora et al definition of using the RAD forced run.

5. My final request would be to ask if you can more directly or relevantly bring this back to complex models – how does this approach help us develop/evaluate/constrain them further? For example you claim in the discussion that the carbon-climate feedbacks are "less sensitive" than the carbon-concentration ones. And that this is due to "the shape of K(Ca,DT)". So how does that help with my ESM? What controls the shape of this in ESMs? And can we measure and constrain it from obs? If so, then your analysis brings a way in which we might narrow the spread in model projections, or at least evaluate a very relevant aspect of the models. If not, then all it does is leave us with a better feel of why the models continue to diverge – if you have any ideas how to make this jump that would be great to see.

Minor comments:

1. Page 4. Your NPP function of CO2 claims to include the effects of climate change – but surely these also depend on the climate sensitivity. For models with high/low climate sensitivity, there is a different trade-off of the effects of CO2 and climate. So I don't follow how the impact on NPP can be made without reference back to the temperature as well as the CO2

2. Page 7. I couldn't quite see if you had a link between ocean heat and ocean carbon uptake – I don't think so. Should these be related? There might be a false extra degree of freedom in your model – I would expect for example rapidly mixed oceans to have high rates of both heat and carbon uptake – and vice versa for poorly mixing oceans. But if you have independent mechanisms of carbon uptake and transient response to climate do you miss this link?

3. Table 1: please be careful to stress "climate sensitivity" as "transient climate response" – you do say so, but using the wrong name makes it look like a very low value (1.8K)

4. Page 14, line 23 – just to check here you mean "1% increase up to double CO2" and not a step-change to double CO2.

---

## Referee Comment (RC2) · M. Heimann (Referee) · 3 Jan 2018

**General comments**

The authors introduce a new variant of a simple analytical, highly parameterised global carbon cycle - climate model, which is used to formally analyse the four major feedback loops in the system, i.e. the land and ocean concentration carbon feedbacks and the land and ocean climate carbon feedbacks. The simplicity of the approach allows the authors to derive analytical approximations to the definitions of various feedbacks metrics at play in the global carbon cycle - climate system.

[Figure]

Simple analytical global carbon cycle models and simple climate models have been used many times in the past. Also the literature contains several simple coupled carbon cycle - climate models (e.g. Gregory et al., 2009 or Meinshausen et al., 2011). It is not clear however, what this particular new variant adds to our understanding of the global carbon cycle - climate system. The motivation outlined in the introduction is not very convincing. The dynamic characteristics of the chosen "mechanistic" model formulation clearly is determined by the simple model structure and the chosen parametrisations of the exchange fluxes. Also the stated "biophysical or biogeochemical interpretation of the model parameters", given that these represent global averages is plausible but not very compelling. E.g. why should the global $CO_2$ fertilisation effect work in reality in a way as parameterized here with a simple $\beta$-factor formulation? Or global respiration with a simple $Q_{10}$ temperature response? Perhaps the main value of the simple model is educational, as it can easily be programmed by students and one can show in this simple model system how the feedback metrics are computed. But as a tool for policymakers nor for generating new carbon cycle science, this model does not provide added value to the already existing simple models. A simple model "tuned" to emulate one or several of the more complex models would be more useful.

Perhaps a missed opportunity for demonstrating the validity of the model is a more careful calibration and evaluation. Clearly the "mechanistic" model parameter values are not based on first principles, but contain large uncertainties. E.g. the $Q_{10}$ value used here (1.72) is highly uncertain (see e.g. Mahecha et al., 2010). Why not tune the model parameters so that the current global carbon budget is properly matched? The model substantially underestimates the historical ocean carbon uptake (Table 2), and, when driven with the historical emissions from the Global Carbon Project (Le Quere et al., 2017), the numerical version of the model underestimates the current ocean uptake. In addition, a graph showing the model performance against the atmospheric $CO_2$ record from ice cores and direct observations could demonstrate that at least on multi-decadal time scales the model performs reasonably. Figure 2 clearly is not sufficient as it does not show any observations. Another useful model evaluation would

be to follow the impulse response simulation protocol defined by Joos et al. (2013) and compare the dynamics of this model with the impulse response simulations of more comprehensive models as shown in that paper.

**Specific comments**

1. As shown in Table 3, the results of the analytical approximations of the feedback metrics compared to the numerical simulations is pretty poor. Does this not invalidate the simplifications made in deriving the analytical approximations?

2. The comparison of the feedback metrics with the results of Zickfeld et al. (2011) and Friedlingstein et al. (2006) in Table 3 shows that the simple model with the chosen parameter values responds substantially different - the discrepancies range up to a factor of 2. This is clearly at odds to what is claimed in section 5.1 and 5.2.

3. On the other hand, also the comprehensive models show a large spread in the feedback metrics. A more useful analysis/comparison would be possible if the model parameters were tuned to emulate the various comprehensive models.

4. The statements in section 5.2 and 5.3 about the behaviour of the carbon cycle - climate system and the feedback metrics under increasing emissions clearly refer to this particular simple model. While plausible, the real world may behave differently.

5. The direct ocean concentration-carbon feedback given as exact in Table A1 and approximated in Table 3 (5th line from bottom) differ very much: Evaluated with the standard model parameters at a value of $ca$ corresponding to 800 ppm the exact formula gives 0.0152 PgC/(ppm yr) while the approximation gives 0.396 PgC/(ppm yr). (I assumed in the exact formula that the symbol $w$ is actually $w_0$).
Also the solid red curve showing $B_O$ in Figure A1a is missing. Obviously there is some error in the listed formulas or the chosen approximation is very poor.

**Technical corrections**

Technically, the formulas in the manuscript contain a some inconsistencies and not correctly defined symbols.

- p. 4, line 25: In the exponent of $Q_R$ the symbol $T$ should be replaced by $\Delta T$.

- p.5, line 13: The way the Revelle factor is used here is weird: Formally, using the notation here, it is defined as:

$$R = \frac{\partial p(c_m, 0)}{\partial c_m} \cdot \frac{c_{m0}}{p(c_{m0}, 0)} \tag{1}$$

Inserting the definition $p(c_m, \Delta T)$ given here (eq (5)) this expression does not evaluate to the constant r as it should according to the text.

- p. 6, line 25: The atmosphere equation, written as an integral equation is weird. Why not write it similar to the biosphere and ocean mixed layer equation as normal first order differential equation?

$$\frac{dc_a}{dt} = e(t) + \frac{Dc_{m0}}{rp(c_{m0}, 0)}(p(c_m, \Delta T) - c_a) + \frac{\mathsf{NPP}_0}{c_{t0}}Q_R^{\Delta T/10}(c_t - K(c_a, T)) + \mathsf{LUC}(t) \tag{2}$$

where $e(t)$ are the emissions (in PgC/yr); E(t) in equation (8) are the integrated emissions (this is nowhere defined in the text, and wrongly described on p.5 line 7).

- p. 7, eq 9: For consistency with the text the symbol $T$ in the differential quotient on the left should be replaced by $\Delta T$.
- Table 3, 4th and 3rd line from bottom: The references to the Figures A1a and A1b are not correct.

- Table A1: What is the meaning of $w$ (without subscript)? Presumably it should be $w_0$?

**Literature**

- Gregory, J. M., Jones, C. D., Cadule, P., and Friedlingstein, P. (2009), Quantifying Carbon Cycle Feedbacks, J Climate, 22(19), 5232-5250.

- Joos, F., Roth, R., Fuglestvedt, J. S., Peters, G. P., Enting, I. G., von, B., W., Brovkin, V., Burke, E. J., Eby, M., Edwards, N. R., Friedrich, T., Frölicher, T. L., Halloran, P. R., Holden, P. B., Jones, C., Kleinen, T., Mackenzie, F. T., Matsumoto, K., Meinshausen, M., Plattner, G.-K., Reisinger, A., Segschneider, J., Shaffer, G., Steinacher, M., Strassmann, K., Tanaka, K., Timmermann, A., and Weaver, A. J. (2013), Carbon dioxide and climate impulse response functions for the computation of greenhouse gas metrics: a multi-model analysis, Atmos Chem Phys, 13(5), 2793-2825.

- Le Quéré, et al. (2017), Global Carbon Budget 2017, Earth System Science Data Discussions, 1-79.

- Mahecha, M. D., Reichstein, M., Carvalhais, N., Lasslop, G., Lange, H., Seneviratne, S. I., Vargas, R., Ammann, C. M., Arain, M. A., Cescatti, A., Janssens, I. A., Migliavacca, M., Montagnani, L., and Richardson, A. D. (2010), Global Convergence in the Temperature Sensitivity of Respiration at Ecosystem Level, Science, 329(5993), 838-840.

- Meinshausen, M., Raper, S. C. B., and Wigley, T. M. L. (2011), Emulating coupled atmosphere-ocean and carbon cycle models with a simpler model, MAGICC6 –
Part 1: Model description and calibration, Atmospheric Chemistry and Physics, 11(4), 1417-1456.

---

## Author Comment (AC1) · 12 Feb 2018

**Response to review by C. D. Jones**

We thank the reviewer for their considered and constructive comments. Please find below our responses to the reviewer's comments. We have uploaded our proposed revised manuscript that addresses this and the other reviewer's comments in a separate Author Comment on the article's Discussion page.

*This is a nicely designed study, and well presented manuscript which attempts to develop and document a simple ("stylized") model of the global climate-carbon cycle system, but in a way which enables analytical analysis of its behaviour and feedback mechanisms. The intended aim is to facilitate improved understanding of the system dynamics and more readily quantify which processes contribute to certain feedbacks and long-term responses.*

*Overall, I very much like the approach and the intention – such modelling studies can develop improved insight and can strip back confounding and complicating issues of model complexity to reveal more fundamental underlying behaviour. I have a few comments below and a few queries about the extent to which the intentions have been realised – I think these can be readily addressed with revised text and some more context/explanation.*

We thank the reviewer for his support.

*Overarching comments:*
*1. There are numerous simple/stylized carbon cycle models in the literature. You cite Raupach (2013) which I like very much. There is also one I developed and have published with several times, including as recently as last year (Jones et al., Tellus, 2003; Jones et al., Tellus, 2006; Jones et al., ERL, 2016). Then there is the MAGICC model often used in IAMs, the Joos IRF, and also the Oscar model which some of the current authors know very well. So I wonder if a bit more explanation is needed for why a new model is required – could you start from one of the existing ones and achieve the same thing? I guess your main driver is the ability to analytically derive the feedback functions – but it's not clear to me the same is not possible from these previous models (I haven't tried it with the Jones et al simple model, but may do!) – Mike Raupach derived eigenmodes, so I would imagine feedback metrics could follow also, but again I haven't tried.*

A key motivation for the proposed model is that it contains mechanistic representations (albeit highly aggregated and stylised) of key climate-carbon processes. In contrast, few, if any, of the models cited above explicitly include a solubility or biological ocean pump. In comparison to many of the cited models, we also substantially simplify the representation of the terrestrial carbon cycle, in order to simplify the analysis. We suspect that a similar analytical feedback analysis could in theory be applied to most of the simple models that the reviewer cites, but the partitioned terrestrial carbon stocks in most of these models would complicate the analysis, and parametric fits to the ocean carbon cycle would make the results less meaningful. We will make clearer the added value of the model in the revised version (see list beginning bottom page 3).

*2. On a similar line – I was looking forward to seeing the analytical derivation of feedback factors, but then realised this was not as "tractable" as your title suggests, and you have to make a lot of assumptions in your sections 4.2 and 4.3. This seems a shame – if this is the case, have you not lost your unique and attractive feature? The resulting expressions are still useful, but not as analytical as you suggest. I also wondered, on seeing the expressions in table 3, if you had compared these to the expressions of Ric Williams et al – who have done a similar based assessment of terms controlling ocean heat/carbon uptake and TCRE. (See e.g., Richard G Williams et al 2016 Environ. Res. Lett. 11 015003)*

We agree with the reviewer that while our feedback results are analytical (in the sense of closed-form mathematical expressions) they are not exact. It would be an interesting challenge to develop a model for which exact results could be achieved, but we suspect this would be at the cost of a mechanistic representation. In the revised version of the manuscript, we will clarify our use of the term of 'analytical'.

Williams et al. (2016) split apart three key factors influencing TCRE: (1) influence of CO2 emissions on radiative forcing from CO2; (2) influence of radiative forcing from CO2 on total radiative forcing; and (3) influence of total radiative forcing on temperature. They then use time series output from ESMs to drive each of these factors and calculate TCRE over time. For example, land and ocean uptake (which influence factor 1) are based directly on ESM output. In contrast, we formulate mechanistic models for land and ocean uptake. Our treatment of factor 3 is similarly highly stylised and we do not explicitly model ocean heat uptake. Regarding factor 2, we only consider CO2 forcing.

In one subsection, Williams et al. analytically calculate an equilibrium TCRE. They use a similar formulation for ocean chemistry based on the Revelle (buffer) factor as ours. However as theirs is an equilibrium calculation, they unlike us do not consider time scales introduced by mechanisms such as the solubility and biological pumps. They also neglect land carbon uptake on this long time scale.

*3. You discuss (page 18, line 16) that you might want to develop this model further to include other mechanisms and forcings. Modellers have attempted a two-box approach for the ocean – by splitting upper and deep ocean before. But I would suggest maybe a two-box approach for the land also – not splitting veg vs soil, but maybe by tropics and extra-tropics as these can have very different responses (even by sign) to changes in climate. IPCC AR5 fig 6.22 shows the latitudinal distribution of "gamma" – there is a clear change of sign towards high latitudes, and so a tropics/high-latitude 2-box approach might be a nice (and novel) development*

To compartmentalise land carbon by tropics and extra-tropics is an interesting suggestion which could be followed in future studies. We will raise this idea in the revised version of the manuscript, see section 6. In doing so, one would define carbon pools not by residence times but by climate sensitivities. This is really interesting but requires some more thinking.

In addition, to our understanding the results in IPCC AR5 fig 6.22 are based on ESM runs that do not contain any representation of permafrost carbon, hence this strong difference

between arctic and extra-arctic beta values seems to more reflect the vegetation response to climate change. It would be interesting to see such sensitivity study using a fully coupled ESM run including permafrost carbon. For now, we therefore refrain from including these sensitivity gradients in our model.

*4. In a few places, you discuss non-linearities – this is good, and important to bring out. It's not necessarily true that land dominates the non-linearity, but this does show up in your rapid-forcing transient runs. If you ran longer, or with slower changing forcing, the ocean would have more chance to exhibit these too – see, e.g. Schwinger et al (J. Climate, 2014). Hence both land and ocean can have pronounced non-linearities and for this reason, C4MIP took the decision to move back towards the Friedlingstein definition of feedbacks as the difference between COUPLED and BGC runs (see Jones et al., GMD, 2016 – documentation of C4MIP). Hence this differs from the Arora et al definition of using the RAD forced run.*

We agree with the reviewer: since we chose to investigate effects on a 100-year policy-relevant time scale, many ocean effects are rendered insignificant. We will discuss in the revised version of the manuscript that the dominance of the land in our nonlinearity results is likely due to the time scale simulated (see section 5.3).

We thank the reviewer for drawing attention to the need for clarity in feedback definitions. Most of the the previous studies to which we compare our numerical feedback results use the Arora definition of feedbacks (Arora 2013 and Zickfeld 2011), the exception is Friedlingstein 2006. We have used the Arora definition to be consistent with the majority of cited previous studies, and will clarify which definition we use in the revised manuscript. We are prepared to also calculate the climate-carbon feedbacks under the Friedlingstein definition if the reviewer wishes, however we feel this would further complicate an already large table.

*5. My final request would be to ask if you can more directly or relevantly bring this back to complex models – how does this approach help us develop/evaluate/constrain them further? For example you claim in the discussion that the carbon-climate feedbacks are "less sensitive" than the carbon-concentration ones. And that this is due to "the shape of K(Ca,DT)". So how does that help with my ESM? What controls the shape of this in ESMs? And can we measure and constrain it from obs? If so, then your analysis brings a way in which we might narrow the spread in model projections, or at least evaluate a very relevant aspect of the models. If not, then all it does is leave us with a better feel of why the models continue to diverge – if you have any ideas how to make this jump that would be great to see.*

We thank the reviewer for this relevant comment. Of course, the divergence amongst ESMs could well be due to diverging parameterisations, as well as different functional forms. As the reviewer suggests, an interesting area for future work would be to study what effects different forms for key functions such as *K* have on feedback strengths. Other steps to aid development of ESMs could include analysing the effective shape of functional forms such

as *K* in ESMs or how to constrain these functional forms from data. These are beyond the scope of the present work but in the revised manuscript we will point to these possible future directions in section 6.

*Minor comments:*
*1. Page 4. Your NPP function of CO2 claims to include the effects of climate change – but surely these also depend on the climate sensitivity. For models with high/low climate sensitivity, there is a different trade-off of the effects of CO2 and climate. So I don't follow how the impact on NPP can be made without reference back to the temperature as well as the CO2*

We agree with the reviewer that an accurate treatment of NPP would separately parameterise the effects of CO2, temperature, rainfall, nutrient availability, and so on. We fold all these effects into a CO2 dependence through Keeling's formula. The references in section 2.1 that we cite for Keeling's formula support this simplification. A key assumption to support this folding, as the reviewer implies, is that we fix climate sensitivity to a constant value -- in the revised manuscript we will state this assumption in section 2.1.

*2. Page 7. I couldn't quite see if you had a link between ocean heat and ocean carbon uptake – I don't think so. Should these be related? There might be a false extra degree of freedom in your model – I would expect for example rapidly mixed oceans to have high rates of both heat and carbon uptake – and vice versa for poorly mixing oceans. But if you have independent mechanisms of carbon uptake and transient response to climate do you miss this link?*

The reviewer is correct that in our model ocean heat uptake (as represented by climate sensitivity) and ocean carbon uptake are parameterised independently. We also agree with the reviewer that higher ocean mixing rates ought to speed up both carbon and heat uptake. We have chosen to focus our model development and analysis on the carbon cycle; future work could involve incorporating mechanisms related to ocean heat uptake such as ocean circulation, and then specifying common drivers on ocean heat and carbon uptake could be worthwhile. We discuss in Section 6 that energy balance is a potential route for further model development.

*3. Table 1: please be careful to stress "climate sensitivity" as "transient climate response" – you do say so, but using the wrong name makes it look like a very low value (1.8K)*

We thank for the reviewer for the cautionary note. We will stress that the climate sensitivity in our model is transient climate response in the revised version (see section 3).

*4. Page 14, line 23 – just to check here you mean "1% increase up to double CO2" and not a step-change to double CO2.*

We thank the reviewer for prompting us to clarify this matter. In fact this simulation has a 1% increase up to *quadrupling* CO2. We will clarify this matter in the revised manuscript (see section 4.3).

---

## Author Comment (AC2) · 12 Feb 2018

Heitzig wrote: "I like this model a lot. You may improve its exposision marginally and make it less time-dependent by introducing a fifth state variable, cd = deep ocean carbon stock, and rewrite eq. (8) as two ODEs, one for cd (containing the two integrals) and one for ca. This way the model's remaining time dependency is only on the two "control" variables E(t) and LUC(t) and it may thus be analysed more easily using tools from bifurcation analysis or topology of sustainable management (see this special issue)."

We thank the commenter for the constructive proposal to improve the readability of the

model and its potential for analysis. In the revised version of the manuscript, we will implement a slightly modified version of the commenter's proposal. We would prefer not to introduce a state variable that corresponds to a quantity (deep ocean carbon) that is outside the boundaries of our system of analysis (which is upper ocean, atmosphere and marine carbon). Instead, we will introduce a new state variable that counts the total amount of carbon over our three carbon stocks. The rate of increase of this quantity will be a differential equation given by the rate of carbon emissions minus the rates of the solubility and biological pumps (Eq. 9 in the revised manuscript). Conservation of carbon within the three internal stocks will then give a simple algebraic equation (Eq. 8 in the revised manuscript) to replace the former Eq. 8.

---

## Author Comment (AC3) · 12 Feb 2018

**Response to review by M. Heimann**

We thank the reviewer for their considered and constructive comments. Please find below our responses to the reviewer's comments. We have uploaded our proposed revised manuscript that addresses this and the other reviewer's comments in a separate Author Comment on the article's Discussion page.

*General comments*
*The authors introduce a new variant of a simple analytical, highly parameterised global carbon cycle - climate model, which is used to formally analyse the four major feedback loops in the system, i.e. the land and ocean concentration carbon feedbacks and the land and ocean climate carbon feedbacks. The simplicity of the approach allows the authors to derive analytical approximations to the definitions of various feedbacks metrics at play in the global carbon cycle - climate system.*

*Simple analytical global carbon cycle models and simple climate models have been used many times in the past. Also the literature contains several simple coupled carbon cycle - climate models (e.g. Gregory et al., 2009 or Meinshausen et al., 2011). It is not clear however, what this particular new variant adds to our understanding of the global carbon cycle - climate system. The motivation outlined in the introduction is not very convincing.*

We thank the reviewer for prompting us to make explicit what our work "adds to our understanding of the global carbon cycle - climate system". As we state in the revised version of the abstract, three specific results of our work are:
- that different feedback formalisms measure fundamentally the same climate-carbon cycle processes;
- that temperature dependence of the solubility pump, biological pump, and CO2 solubility all contribute approximately equally to the ocean climate-carbon feedback; and
- that concentration-carbon feedbacks may be more sensitive to future climate change than climate-carbon feedbacks.

These results would not have been possible without the simple, mechanistically-based model that we develop in this manuscript.

Regarding the previous models that the reviewer cites: There is no explicit representation of biophysical processes in the model of Gregory et al. (2009), which consists of fits to fluxes between ocean, atmosphere and land carbon stocks predicted by C4MIP models, or the ocean carbon cycle component of Meinshausen et al. (2011), which is a parametric fit to predicted impulse response functions. That our model is a mechanistically based representation of the carbon cycle, even if that representation is highly aggregated and simplified, allows us to deliver the insights above. We concede however that our motivation of the model in the Introduction was not particularly detailed on these points. In the revised manuscript we will expand upon the model's motivation in the second paragraph of the Introduction, as well as refining the more detailed description at the start of section 2.

*The dynamic characteristics of the chosen "mechanistic" model formulation clearly is determined by the simple model structure and the chosen parametrisations of the exchange fluxes. Also the stated "biophysical or biogeochemical interpretation of the model parameters", given that these represent global averages is plausible but not very compelling. E.g. why should the global CO2 fertilisation effect work in reality in a way as parameterized here with a simple β-factor formulation? Or global respiration with a simple Q10 temperature response?*

We use the term "mechanistic" to convey that we have in our model representations of real-world processes, such as photosynthesis, respiration, ocean-atmosphere diffusion and the solubility and biological pumps. Our model is not a precise, first-principles mechanistic description of these processes at the microscopic scale, but then again all models are simplifications of reality; we merely choose to perform the simplification at a more aggregated level than most Earth System Models. The β-factor (or 'Keeling formula') and Q10 temperature responses are previously used parameterisations of the response to climate change of globally aggregated NPP and respiration, respectively. We will clarify our use of the term 'mechanistic' in the revised manuscript (see second paragraph of the introduction).

*Perhaps the main value of the simple model is educational, as it can easily be programmed by students and one can show in this simple model system how the feedback metrics are computed. But as a tool for policymakers nor for generating new carbon cycle science, this model does not provide added value to the already existing simple models. A simple model "tuned" to emulate one or several of the more complex models would be more useful.*

Gregory et al. (2009) and Meinshausen et al. (2011), as well as others, already provide simple models '"tuned" to emulate one or several of the more comprehensive models. We believe there is scope for a model such as ours, in which we do not force our model to closely fit historical data (or future projections) but rather parameterise each process with the best available (globally aggregated) knowledge about that process. See our response to the next comment below for further information.

*Perhaps a missed opportunity for demonstrating the validity of the model is a more careful calibration and evaluation. Clearly the "mechanistic" model parameter values are not based on first principles, but contain large uncertainties. E.g. the Q10 value used here (1.72) is highly uncertain (see e.g. Mahecha et al., 2010). Why not tune the model parameters so that the current global carbon budget is properly matched? The model substantially underestimates the historical ocean carbon uptake (Table 2), and, when driven with the historical emissions from the Global Carbon Project (Le Quere et al., 2017), the numerical version of the model underestimates the current ocean uptake. In addition, a graph showing the model performance against the atmospheric CO2 record from ice cores and direct observations could demonstrate that at least on multi-decadal time scales the model performs reasonably. Figure 2 clearly is not sufficient as it does not show any observations. Another useful model evaluation would be to follow the impulse response simulation protocol defined by Joos et al. (2013) and compare the dynamics of this model with the impulse response simulations of more comprehensive models as shown in that paper.*

We thank the reviewer for these comments and suggestions on model calibration and evaluation. Following the reviewer's suggestion, in the revised version of the paper we will include historical carbon fluxes and temperature anomalies alongside model predictions (see revised Fig 2).

We have attempted to 'tune' several different combinations of parameters to match current carbon stocks (one example is $K_C = 0.25$, $Q_R = 2.5$ and $w_0 = 0.2$). However the tuned parameter sets lie well outside the best available independent estimates of those parameters (see references in Table 1). This is not surprising since we do not expect a mechanistically based of model of this simplicity to precisely reproduce historical carbon stocks.

Rather than forcing the model to fit historical data, we choose to parameterise each process with the best available knowledge about that process. Gaps between our model and observations then point to what other processes should be included in a more complex model to improve accuracy. This is in line with our stated model purposes of understanding and learning, rather than emulation and prediction. In the revised manuscript, we will clarify our choices taken during the parameterisation of our model (see section 3).

*Specific comments*
*1. As shown in Table 3, the results of the analytical approximations of the feedback metrics compared to the numerical simulations is pretty poor. Does this not invalidate the simplifications made in deriving the analytical approximations?*

We concede that in the submitted version of the manuscript, while the land feedback metrics were accurate, the agreement between the numerical and analytical results for ocean feedback metrics was poor. Deriving approximate metrics for ocean feedbacks is challenging, as the deep ocean does not reach equilibrium on the time scale of our simulation. We have taken the opportunity to derive alternative approximations to the ocean feedback metrics (see description in section 4.2). The approximated ocean climate-carbon feedback is now more accurate. The ocean concentration-carbon feedback remains in poor agreement. As we will explain in the revised manuscript (see second paragraph of section 5.2), this is partly due to an approximation made in analytically estimating the deep ocean uptake, but partly also due to numerical concentration-carbon feedback calculations requiring climate-carbon feedbacks to be switched off.

*2. The comparison of the feedback metrics with the results of Zickfeld et al. (2011) and Friedlingstein et al. (2006) in Table 3 shows that the simple model with the chosen parameter values responds substantially different - the discrepancies range up to a factor of 2. This is clearly at odds to what is claimed in section 5.1 and 5.2.*

We thank the reviewer for prompting us to clarify what we judge as 'agreement' between the results of our simple model and the results of previous simulations. First, we note that the results of complex models display considerable spread (as also noted by the reviewer in the following point below). While some of our results differ by nearly a factor of 2 from the mean results of Friedlingstein et al, all our feedback metric results are within their reported spread

(Table 3). Second, we consider it remarkable that such a simple model can reproduce the results of highly complex models so closely, and would not consider a discrepancy of a factor of 2 an invalidation of the simple model. We will discuss these discrepancies more carefully in the revised manuscript (see first paragraph of section 5.2).

*3. On the other hand, also the comprehensive models show a large spread in the feedback metrics. A more useful analysis/comparison would be possible if the model parameters were tuned to emulate the various comprehensive models.*

This is an interesting idea, but beyond the scope of our study. As discussed above, our goal is not to emulate or evaluate ESMs, but rather to develop process-based understanding.

*4. The statements in section 5.2 and 5.3 about the behaviour of the carbon cycle - climate system and the feedback metrics under increasing emissions clearly refer to this particular simple model. While plausible, the real world may behave differently.*

We thank the reviewer for raising this concern. It is correct that our model can only anticipate changes in the carbon cycle arising from those processes that it has modelled -- and may therefore neglect other important future changes in the carbon cycle. We will acknowledge this caveat in the manuscript (see second-last paragraph of section 5.2).

*5. The direct ocean concentration-carbon feedback given as exact in Table A1 and approximated in Table 3 (5th line from bottom) differ very much: Evaluated with the standard model parameters at a value of ca corresponding to 800 ppm the exact formula gives 0.0152 PgC/(ppm yr) while the approximation gives 0.396 PgC/(ppm yr). (I assumed in the exact formula that the symbol w is actually w0). Also the solid red curve showing BO in Figure A1a is missing. Obviously there is some error in the listed formulas or the chosen approximation is very poor.*

We respectfully disagree with the reviewer's calculations. By our calculations, under the conditions the reviewer indicated the approximation gives 0.0398 PgC/(ppm yr). While this is not as severe as the 20-30 times the reviewer suggested, it is still a significant difference at 2 to 3 times the exact expression. We took the opportunity to derive a more precise approximation (see Table 3 and the last sentence of section 4.3) that gives a value 0.0240 PgC/(ppm yr).

We thank the reviewer for noticing the omitted curve in Figure A1; this will be rectified in the revised manuscript (see our proposed revision).

*Technical corrections*
*Technically, the formulas in the manuscript contain a some inconsistencies and not correctly defined symbols.*
*• p. 4, line 25: In the exponent of QR the symbol T should be replaced by $\Delta T$.*

Thank you, in the revised manuscript we will correct this mistake.

*• p.5, line 13: The way the Revelle factor is used here is weird: Formally, using the notation here, it is defined as:*

$$R = \frac{\partial p(c_m, 0)}{\partial c_m} \frac{c_{m0}}{p(c_{m0}, 0)}$$

*Inserting the definition p(cm, ∆T) given here (eq (5)) this expression does not evaluate to the constant r as it should according to the text.*

We respectfully disagree with the reviewer's general definition of Revelle factor. According to Sabine et al. (2004) [see citation in our manuscript] and the AR4 [see https://www.ipcc.ch/publications_and_data/ar4/wg1/en/ch7s7-3-4-2.html], the general definition of Revelle factor is in our notation

$$R = \frac{\partial p(c_m, 0)}{\partial c_m} \frac{c_m}{p(c_m, 0)}$$

that is, the mixed-layer ocean carbon stock in the right-hand quotient should not be fixed at pre-industrial $CO_2$ levels. Substituting our model's expression for partial pressure of $CO_2$ [equation 5 in our manuscript] gives R = r for all $c_m > c_{m0}$ as expected.

*• p. 6, line 25: The atmosphere equation, written as an integral equation is weird. Why not write it similar to the biosphere and ocean mixed layer equation as normal first order differential equation?*
*<equation omitted>*
*where e(t) are the emissions (in PgC/yr); E(t) in equation (8) are the integrated emissions (this is nowhere defined in the text, and wrongly described on p.5 line 7).*

We agree that the form of the atmosphere equation is unusual! In line with the suggestion of the comments provided by Heitzig (see above), we will rewrite equation 8 as an algebraic equation for conservation of carbon amongst our stocks, alongside a new differential equation to account for aggregate carbon flows into or out of our three stocks. This formulation will remove all integral equations.

We thank the reviewer for identifying that E(t) is incorrectly defined. We will correct this mistake in the revised manuscript (see definition preceding the new equation 9 and section 3).

*• p. 7, eq 9: For consistency with the text the symbol T in the differential quotient on the left should be replaced by ∆T.*

Thank you, in the revised manuscript we will make this change to improve the clarity of the manuscript (see equation 10 in the revised manuscript).

*• Table 3, 4th and 3rd line from bottom: The references to the Figures A1a and A1b are not correct.*

Thank you for noting this mistake. We will correct the figure numbering in the revised manuscript.

• *Table A1: What is the meaning of w (without subscript)? Presumably it should be w0?*

We thank the reviewer for noting this mistake. We confirm the w in Table A1 should be w0. We will correct this mistake in the revised manuscript.

---

## Author Comment (AC5) · 12 Feb 2018

We thank the reviewers for their considered and constructive comments that have improved the manuscript.

Reviewer Jones wrote "[t]his is a nicely designed study, and well presented manuscript" and "I very much like the approach and the intention". In line with the reviewer's suggestions, we now compare our model against additional existing models, clarify our use of the terms "mechanistic" and "analytical", include additional speculations on future work, and responded to some specific requests for clarification.

[Figure]

Reviewer Heimann raised concerns about the novelty of our model compared to previous models and about how accurately our numerical and analytical results compare to previous ESM results as well as to each other. We believe there exists a gap in the literature for our mechanistically-based but highly stylized model, as we have sought to make clearer in the revised manuscript. For a model as extremely simple as ours, and given the wide spread in the results of ESMs, we believe our results are sufficiently accurate. In line with Heimann's very perceptive technical corrections we have modified our feedback calculations to bring them into closer agreement than in the initially submitted version.

We also appreciate Heitzig's comment to modify the mathematical presentation of the model. We have modified the model to achieve Heitzig's overall goal of presenting the model solely in terms of differential, instead of integro-differential, equations, although not exactly by the route he suggested.

Please find attached our proposed revised manuscript, with changes tracked.

Please also note the supplement to this comment:
https://www.earth-syst-dynam-discuss.net/esd-2017-78/esd-2017-78-AC5-supplement.pdf

**Supplement:**

[revised manuscript text omitted]

---

## Author Response (AR2)

**RESPONSE TO REVIEWERS -- SECOND RESUBMISSION**

We again thank Chris Jones for his review of our manuscript. Please find below our responses to his comments and a description of how we changed the manuscript accordingly. We also made a small number of edits to improve the clarity of the manuscript. All changes are marked in the version of the manuscript provided below.

*1.        Multiple review comments asked - is there a need for a new simple model? OK, the authors make a good argument for this, and I accept it fits a niche - the ability to understand the simplicity of it is nice. I still feel that the aspect of "analytically" solving the behaviour of the model is a little over-sold given that further simplifying assumptions have to be made, but nonetheless the usefulness is clear.*

We thank the reviewer for their comment. We infer no required action.

*2.        I accept that the table of feedbacks is already full and so there is no need to add the Friedlingstein definitions of feedback metrics to it.*

We thank the reviewer for their comment. We infer no required action.

*3.        I would like to see some discussion of the implications of not tuning parameters to optimise the historical performance against observations. I certainly understand the argument that using best estimates of parameters keeps the model closer in a "process sense" to reality. But to some extent you cannot pick and choose when reality is a good test - for example in your response to Martin Heimann you say that you don't see "a discrepancy of a factor of 2 an invalidation of the simple model", but then your first sentence in the conclusions highlights that this model emulates the historical record - so either you see this as important or you don't! Perhaps a good way forward is to accept your parameter choice, but can you show that your main conclusions are not dependent on this - i.e. if you tuned the model to get closer to the historical record do you retain or lose your main conclusions? If you are correct then they should be robust... If they are dependent on tuning and that getting closer to historical global behaviour breaks your conclusions then to what extent can we really believe them?*

We thank the reviewer for their constructive comment. As suggested, we performed an additional simulation with parameters fitted to historical changes (see Table A2). We found our results regarding agreement with predicted future carbon stock changes and feedback estimates to be largely unchanged, or at most slightly worse (see new text in section 5.1 and a new third paragraph in section 5.2). Our other results are based on the qualitative analytical forms and are not affected by changes in parameter values.

*END COMMENTS*

[revised manuscript text omitted]